# Exploring a Bioequivalence Failure for Silodosin Products Due to Disintegrant Excipients

**DOI:** 10.3390/pharmaceutics14122565

**Published:** 2022-11-23

**Authors:** Isabel González-Álvarez, Bárbara Sánchez-Dengra, Raquel Rodriguez-Galvez, Alejandro Ruiz-Picazo, Marta González-Álvarez, Alfredo García-Arieta, Marival Bermejo

**Affiliations:** 1Engineering: Pharmacokinetics and Pharmaceutical Technology Area, Miguel Hernandez University, 03550 San Juan de Alicante, Spain; 2Service of Pharmacokinetics and Generic Medicines, Division of Pharmacology and Clinical Evaluation, Department of Human Use Medicines, Spanish Agency for Medicines and Health Care Products, 28022 Madrid, Spain

**Keywords:** Biopharmaceutics Classification System (BCS), pharmacokinetics (PK), in vitro dissolution, permeability, bioequivalence

## Abstract

Some years ago, excipients were considered inert substances irrelevant in the absorption process. However, years of study have demonstrated that this belief is not always true. In this study, the reasons for a bioequivalence failure between two formulations of silodosin are investigated. Silodosin is a class III drug according to the Biopharmaceutics Classification System, which has been experimentally proven by means of solubility and permeability experiments. Dissolution tests have been performed to identify conditions concordant with the non-bioequivalent result obtained from the human bioequivalence study and it has been observed that paddles at 50 rpm are able to detect inconsistent differences between formulations at pH 4.5 and pH 6.8 (which baskets at 100 rpm are not able to do), whereas the GIS detects differences at the acidic pH of the stomach. It has also been observed that the differences in excipients between products did not affect the disintegration process, but disintegrants did alter the permeability of silodosin through the gastrointestinal barrier. Crospovidone and povidone, both derivatives of PVP, are used as disintegrants in the test product, instead of the pregelatinized corn starch used in the reference product. Permeability experiments show that PVP increases the absorption of silodosin—an increase that would explain the greater C_max_ observed for the test product in the bioequivalence study.

## 1. Introduction

A generic product is a medicine that is developed to be equivalent and interchangeable with a reference medicinal product that has been previously authorized. In all cases, generic products must meet the following conditions: to contain the same active ingredient, with the same strength and the same dosage form for the same route of administration. In addition, generic products are authorized only after the exclusivity period of the reference product, which is at least 10 years in the European Union, and only after the patent and its complementary certificate have expired. Finally, generic products must demonstrate to be bioequivalent to the reference product [1,2].

Generic products meet the same quality standards as any other medicinal product, and support their safety and efficacy profile by demonstrating bioequivalence to the reference product. Generic products are commercialized with a lower price than the original product because their developmental process is much shorter and cheaper, which contributes to a more rational use of economic resources in the health system [3].

Normally, bioequivalence is evaluated in humans employing a randomized, two-period, two-sequence, single-dose crossover design. In these studies, plasma levels of the parent drug are measured in different participants and the rate and extent of absorption of the drug present in each product is compared by means of the primary pharmacokinetic parameters C_max_ (maximum plasma concentration or peak exposure) and AUC (the area under the concentration time curve), which reflect the absorption rate and the extent of exposure, respectively [4]. Two formulations will be considered bioequivalent, and thus, with a comparable in vivo performance, if the 90% confidence intervals of the ratio of the geometric means of test and reference for both Cmax and AUC are between 0.80 and 1.25, which ensures a difference equal to or lower than 20% between test and reference and is considered an irrelevant difference that does not require an adjustment of the dose [5,6].

According to Gordon Amidon et al., the main processes that affect absorption are the aqueous solubility of the active substance and its permeability through the intestinal barrier. Taking these processes into account, they defined the Biopharmaceutics Classification System (BCS) and divided drugs into four different groups: (I) high permeability and high solubility, (II) high permeability and low solubility, (III) low permeability and high solubility, and (IV) low permeability and low solubility [7,8].

The main purpose of the BCS is to reduce the number of generic drugs that must demonstrate their bioequivalence by means of studies in humans and, therefore, reduce ethical and economic difficulties in the development of these types of products [9]. In line with this idea, governmental agencies unified criteria in July 2020 and defined a drug as highly permeable when its oral fraction absorbed is equal to or greater than 85%, and highly soluble when its highest single therapeutic dose is completely soluble in 250 mL of aqueous media over the pH range of 1.2–6.8 at 37 ± 1 °C. (10) Highly soluble drugs (class I and class III) whose transport through the intestinal barrier follows a passive route can be candidates for a biowaiver and, thus, demonstrate similarity between formulations by means of in vitro dissolution tests [10].

Silodosin is a drug whose commercialization was first approved in 2008 by the U.S. Food and Drug Administration (FDA) and in 2010 by the European Medicines Agency (EMA). It is indicated for the treatment of benign prostatic hyperplasia as it allows a better urine flow. In terms of physicochemical properties, silodosin is a weak base with a pKa equal to 9.66 and a logP of 3.05. It is slightly soluble in water and, according to the literature, it is classified as class III [11,12,13]. Nevertheless, it could not be a candidate for a BCS biowaiver as studies in Caco-2 cells have demonstrated it is a substrate of P-glycoprotein (Pgp) and of active absorption processes through MRP3 [14,15].

The objectives of this work were to confirm the BCS classification of silodosin and to explain the reasons for the different bioavailability, which caused a bioequivalence failure, between a new generic product of silodosin and its reference product. With this aim, solubility, disintegration, dissolution and permeability were studied.

## 2. Materials and Methods

### 2.1. Drugs and Products

Reference product (Silodyx^®^) was acquired from a local pharmacy. Test product and pure silodosin were kindly supplied by a pharmaceutical company. Reference and test products were capsules containing 4 mg of silodosin and their different excipients as shown in Table 1.

Sodium chloride, sodium acetate, potassium dihydrogen phosphate, hydrochloric acid, acetic acid, phosphoric acid, methanol, trifluoroacetic acid (TFA), rhodamine, and polyvinylpyrrolidone (PVP) were purchased from Sigma^®^ (Barcelona, Spain).

### 2.2. In Vivo Study

The in vivo study (EudraCT no.: 2017-001854-32) was an open label, balanced, randomized, two-period crossover BE study in 48 healthy subjects. The volunteers received one immediate release (IR) dose of the test product and one dose of the reference product in a sequence determined by randomization. Maximum plasma concentration (C_max_) and the area under the concentration time curve (AUC_0-t_) were the primary pharmacokinetic parameters to determine the bioequivalence between the products.

The results of the BE study are described in Table 2, where the confidence intervals for C_max_ and AUC_0-t_ showed the BE failure due to the C_max_ value, whose 90% confidence interval upper boundary was too high. More importantly, as the confidence interval does not include the 100% value, a statistically significant difference at the α level of confidence interval was detected between the test and the reference products. This unexpected difference caused the marginal bioequivalence failure.

### 2.3. Solubility Assays: Saturation Shake-Flask Procedure

Silodosin solubility was determined at pH 1.2, 4.5 and 6.8 using an orbital shaker at 100 rpm and 37 °C. First, a preliminary test was performed to determine the time needed to reach an equilibrium concentration, for which 33 mg of API and 2 mL of buffer were used. Once the equilibrium time was known, a final test with 1 mg of API and 5 mL of buffer was performed in which equilibrium solubility at pH 1.2, 4.5 and 6.8 was measured by HPLC. Standard buffers were prepared according to European Pharmacopeia: pH 1.2 (Sodium chloride 50 mM), pH 4.5 (Sodium acetate 36.5 mM), and pH 6.8 (Potassium dihydrogen phosphate 50 mM).

### 2.4. Disintegration Assays

Disintegration assays were carried out in a PTZ-S disintegration tester (Pharma Test^®^, Hainburg, Germany) with 900 mL of buffer at pH 1.2 or 900 mL of water. Temperature was fixed to 37.3 °C, and the oscillations per minute of the basket to 30 times per minute. The times at which the formulations began to disintegrate were noted, as well as the times at which they were completely disintegrated. Statistical differences were evaluated with software SPSS (V 21.0) assuming a statistical level of 0.05.

### 2.5. Dissolution Assays: USP I and USP II

Several dissolution tests were performed at pH 1.2, 4.5 and 6.8 with a PT-DT70 dissolution instrument (Pharma Test^®^). In all cases, the temperature was fixed to 37 °C, the volume of buffer employed to 900 mL, and the samples times to 5, 10, 15, 20, 30, 45, and 60 min. The apparatus and the revolutions per minute employed were:Paddle apparatus with sinkers to prevent capsules from floating: 50 rpm.Basket apparatus: 100 rpm. The assays with baskets at 100 rpm were performed by the pharmaceutical company and the sampling times were 7.5, 15, 20, and 30 min.

Dissolution profiles were compared using the similarity factor f_2_ (Equation (1)), in which n is the number of points, R(t) and T(t) are the mean percent of reference or test product dissolved at time t. Two profiles are considered similar when their f_2_ value is ≥50 or when, in both products, the fraction dissolved is ≥85% in 15 min, in which case it is not necessary to calculate f_2_ [10].
(1)$$f_{2}=50\times \log\frac{100}{\sqrt{\left(1+\left(\frac{\sum_{t=1}^{n}{\left[\mathrm{Rt}-\mathrm{Tt}\right]}^{2}}{n}\right)\right)}}$$

### 2.6. Permeability Assay: Doluisio Experiment

The product permeabilities were evaluated in rats using the in situ closed loop perfusion experiment (Doluisio technique) in three different segments of the intestinal tract (duodenum, jejunum, and ileum) [16,17]. The animal experiments, performed in 250–300 g male Wistar rats, were designed according to the document approved by the Spanish Government with code A1330354541263. With this aim, three loops were made in each rat and capsules were opened and resuspended in 250 mL of buffer at pH 6.8. Then, 2 mL were administered in the duodenum segment, 4 mL in the jejunum one and 4 mL in the ileum one. Samples were taken at 5, 10, 15, 20, 25, and 30 min and, after being centrifuged, were analyzed by HPLC.

Once the samples were analyzed, the apparent absorption rate (k_app_) and the apparent permeability (P_app_) were calculated with Equations (2) and (3), in which C is the luminal concentration of silodosin at sampling times after correcting it by water reabsorption, C_0_ is the extrapolated concentration of silodosin at time 0, and R is the radius of the intestinal segment [18]. Statistical differences were evaluated with software SPSS (V 21.0) assuming a statistical significance level of 0.05.
(2)$$C=C_{0}\cdot e^{-k_{\mathrm{app}}\cdot t}$$
(3)$$P_{\mathrm{app}}=k_{\mathrm{app}}\cdot \frac{R}{2}$$

The Doluisio studies were approved by the Scientific Committee of the Faculty of Pharmacy, Miguel Hernandez University, and followed the guidelines described in the EC Directive 86/609, the Council of the Europe Convention ETS 123, and Spanish national laws governing the use of animals in research.

### 2.7. In Vitro Permeability Tests

The permeability of the API and the API with crospovidone, povidone, or pregelatinized corn starch was evaluated in MDCK-MDR1 monolayers, a cell line which simulates the intestinal barrier and expresses the P-gp [19].

First, cells were seeded in inserts in 6-well plates and maintained over 7 days as performed in Sanchez-Dengra et al. [20]. 

After defrosting the cells, the passage numbers between 30 and 40 were used to perform the experiments. These cells were maintained in media to grow the cells in the wells.

The medium used to maintain the growth of the cells was DMEM (Dulbecco’s Modified Eagle’s Medium with 4500 mg/L glucose, l-glutamine sodium bicarbonate, without sodium pyruvate from Sigma D5796) (89%) and it was combined with MEM Non-Essential Amino Acids Solution from Gibco 11140–035 (1%), Fetal Bovine Serum F7524 from Sigma (10%), and HEPES 1 M 15630-056 by Gibco.

The media were changed three times each week. On the day of the experiment, the apical side of the wells were filled with the drug solutions so that the Apical-to-Basolateral effective permeability—P_eff_A-B could be determined. Samples were taken from the basolateral side at 15, 30, 60, and 90 min. The cell culture is considered acceptable for transport experiments if all the following criteria are met:(a)TEER values at the beginning and at the end of the permeability experiments should be adequate (100 units of difference with blank insert and no more than a 10% difference between initial values and values at the end of the experiment).(b)In checking the mass balance after determining the amount of compound in the insert membranes and inside the cells, the percentage of compound retained in the cell compartment should typically be less than 5%.

Later on, the samples were analyzed with an HPLC and the apparent permeability values were calculated using the Modified Non-Sink equation [21].

### 2.8. Dissolution Experiments: Gastrointestinal Simulator (GIS)

The gastrointestinal simulator is a new device which comprises three dissolution compartments: (i) a gastric chamber (GIS Stomach), (ii) a duodenal chamber (GIS Duodenum) and (iii) a jejunal chamber (GIS Jejunum); it is controlled by a computer. The design of the GIS is depicted in Figure 1.

A tablet of each product was added to the stomach compartment at the start of the study. The dissolution media, initial volumes, and secretion rates are described in Table 3.

Gastric emptying was programmed as a first-order kinetic process with a gastric half-life of 13 min, to mimic the human values reported in the literature [22]. Duodenal volume must remain constant, therefore, the pump which connects the duodenum and the jejunal compartment should be modulated accordingly in order to achieve this requirement. The volume in duodenum was 50 mL. The volume in the jejunal compartment was 0 at the beginning of the experiment and this compartment was an accumulation of all of the experiments. All compartments were stirred with a paddle system (Muscle Corp., Osaka, Japan) which allowed for a stir rate of 20 rotations per minute (RPM).

The fluids were transferred from the stomach to the duodenum and from the duodenum to the jejunum by peristaltic pumps (Ismatec REGLO pump; IDEX Health and Science, Glattbrugg, Switzerland).

Samples were collected at different times and they were centrifuged and diluted with methanol. After that, samples were analyzed by HPLC.

### 2.9. HPLC Analysis

Samples from the solubility, dissolution, and permeability assays were analyzed by HPLC employing a UV detector (Waters^®^ 2487, Milford, MA, USA), an X-Bridge^®^ C18 column (3.5 μm, 4.6 × 100 mm), and a mobile phase of 50% methanol and 50% acid water (0.05% *v*/*v* TFA in water). The wavelength was set to 225 nm, the flow of the mobile phase to 1 mL/min, and the temperature to 30 °C. Under these conditions, the retention time for silodosin was determined to be 5 min. The lowest limit of detection of silodosin was 0.189 μg/mL and the lowest limit of quantification of silodosin was 0.631 μg/mL.

## 3. Results

### 3.1. Solubility Experiments: Saturation Shake-Flask Procedure

Figure 2 shows the experimental solubility of silodosin obtained at pH 1.2, 4.5, and 6.8. It can be observed that solubility decreases with pH; however, as the maximum therapeutic dose of silodosin is 8 mg (8/250 = 0.032 mg/mL), it can be classified as a high-solubility drug.

### 3.2. Disintegration Tests

The disintegration times for the reference product and the test product were not statistically different as shown in Figure 3. At pH 1.2, both the beginning of the disintegration and the completed disintegration process were faster than with water, but in all cases, the disintegration rate for both formulations was the same.

### 3.3. Dissolution Tests: USP I and USP II apparatuses

Figure 4 shows the results for the different dissolution tests that were carried out with baskets and paddles, and Table 4 shows the similarity results after comparing the different profiles with the similarity factor f_2_. Only the experiments with paddles at 50 rpm (pH 4.5 and 6.8) are able to detect the dissimilarity between products in accordance with the in vivo bioequivalence study.

### 3.4. Permeability Assay: Doluisio’s Experimental Technique

Permeability values of the products containing silodosin are shown in Figure 5. It can be observed that, for the reference product, the permeability gets lower while it passes from the duodenum to the jejunum and then to the ileum, while for the test product, the permeability keeps constant throughout the intestinal tract.

### 3.5. In Vitro Permeability Tests

In Figure 6, it can be observed how the use of crospovidone as a disintegrant, at the concentration present in the test formulation, significantly increases the permeability of silodosin in regard to the API on its own and the API with povidone and pregelatinized corn starch.

### 3.6. Dissolution Experiments: Gastrointestinal Simulator (GIS)

The use of a more complex dissolution apparatus, the GIS, demonstrated the ability to show the lack of similarity between the test and the reference product in the stomach compartment, as shown in Figure 7. Nonetheless, differences are not maintained in the other compartments.

## 4. Discussion

This study confirms the BCS classification of silodosin and explains that the P-gp inhibition caused by crospovidone is the most likely reason for the bioequivalence failure (i.e., non-bioequivalence) of a new generic product of silodosin with respect to its reference product (Silodyx^®^), because, as it can be observed from the results of the human bioequivalence study (Table 2), the upper boundary of the 90% confidence interval of C_max_ was 126.08%, which is marginally above the upper limit for bioequivalence (1.25).

Solubility assays have confirmed that silodosin can be considered a drug with high solubility as its maximum therapeutic dose (8 mg) can be freely dissolved in 250 mL of buffer at pH 1.2, 4.5, and 6.8 (Figure 2). In addition, according to the API permeability assays (Figure 6), silodosin can be considered a drug with low permeability, as its permeability coefficient in MDCK-MDR1 (2.38 × 10^−5^ cm/s) is lower than that of metoprolol in the same cell line (6.77 × 10^−5^cm/s) [23], an metoprolol is the model drug used to classify permeability [8,24]. Therefore, it has been confirmed that silodosin is a BCS class III drug.

According to Table 1, the reference and test capsules differ in their excipient composition: pregelatinized corn starch in the reference was replaced by crospovidone and by povidone in the test product, as all of them are commonly used as disintegrants [25,26,27,28]. Povidone and crospovidone are obtained from PVP and, depending on the degree of polymerization, they can be classified as soluble PVP (povidone) or insoluble PVP (crospovidone) [26]. In terms of disintegration, Figure 3 shows that the change was not relevant as there are no differences in the disintegration process between the reference product and the test product.

In class III drugs, the dissolution process tends to be less important and, in fact, from the results obtained in the USP I and the USP II, shown in Table 4 and Figure 4, the only conditions that are able to simulate the non-bioequivalence observed in vivo are the experiments with paddles at 50 rpm (6.8). Importantly, at pH 4.5, the dissolution profiles are in the reverse order. The reference product dissolves more rapidly, which illustrates that the results of the paddle apparatus are not consistent and, therefore, unreliable. The M9 ICH guideline on the BCS biowaivers proposes the use of the paddle apparatus at 50 rpm or the basket apparatus at 100 rpm, indistinctly, for evaluating the bioequivalence in vitro [10]. Nevertheless, according to the results presented here, it is not the same to use baskets at 100 rpm as it is to use paddles at 50 rpm. As a matter of fact, the pharmaceutical company that failed in the development of the generic drug carried out dissolution tests with baskets at 100 rpm, from which they concluded that both formulations were similar. However, had they used paddles, they would have observed that the formulations were not similar and, therefore, could have made some changes before going into the bioequivalence study in humans. Even if these dissolution tests are not entirely predictive, it would have been advisable to show similar dissolution profiles before conducting expensive bioequivalence studies in humans.

The dissolution experiments carried out with the GIS show that a more complex dissolution method is also able to detect the differences between the reference product and the test product, as it can also be observed in the stomach compartment (Figure 7). In this apparatus, in which the stomach compartment has a lower volume compared with the USP apparatus and the hydrodynamic conditions better mimic the human stomach, it is possible to observe the difference in disintegration and dissolution. Dissolution is similar in the next segments, but a faster dissolution in the stomach combined with better permeability in the intestines may give the test formulation an advantage for its absorption rate. Although, unless an absorption window in the duodenum exists, the limited permeability of silodosin would make the dissolution difference in the stomach irrelevant. The permeability differences are considered the more relevant factor for explaining the difference in C_max_ for low-permeability drugs without an absorption window in the duodenum.

Some years ago, excipients were considered inert substances irrelevant in the absorption process. However, years of study have demonstrated that this belief is not always true [29]. Silodosin transport across the intestinal membrane is a combined transport of passive diffusion and active efflux transport through Pgp and active influx transport through MRP3. Thus, an interaction between the transporters and any of the excipients could alter silodosin permeability [14,15].

Reference and test formulations were tested and their permeability experiments show that the absorption of the reference product decreases while it advances through the gastrointestinal tract [30]. The test formulation contained crospovidone and povidone and its permeability values remained constant, which shows that povidone and crospovidone could inhibit Pgp or activate MRP3 (Figure 5).

This inhibition has been confirmed in the in vitro permeability study as shown in Figure 6. In addition, it has been observed that the disintegrant that is responsible for the failure in the bioequivalence study is crospovidone seeing as the permeability of the test drug increases considerably in the presence of crospovidone, whereas its permeability does not increase significantly with the API alone, the API with povidone, or with the pregelatinized corn starch.

## 5. Conclusions

The BCS classification of silodosin (class III) has been experimentally confirmed in this study. The main reason for the borderline bioequivalence failure, due to a small difference in C_max_, was the use in the test product of crospovidone as a disintegrant instead of pregelatinized corn starch as was used in the reference product. It can be concluded that the selection of the correct excipients is a key step in the development of new generic products. It is advisable not to change the excipients in the case of the BCS class III drugs, unless it is mandatory due to patent protection, since knowing how the excipients affect the release from the product is as important as knowing their potential effects on the gastrointestinal system and its functions.

## Figures and Tables

**Figure 1 pharmaceutics-14-02565-f001:**
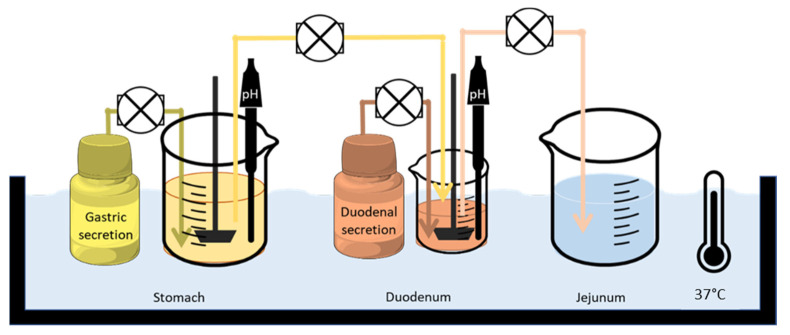
Setup and design of the GIS that was applied to test the products in fasted-state conditions.

**Figure 2 pharmaceutics-14-02565-f002:**
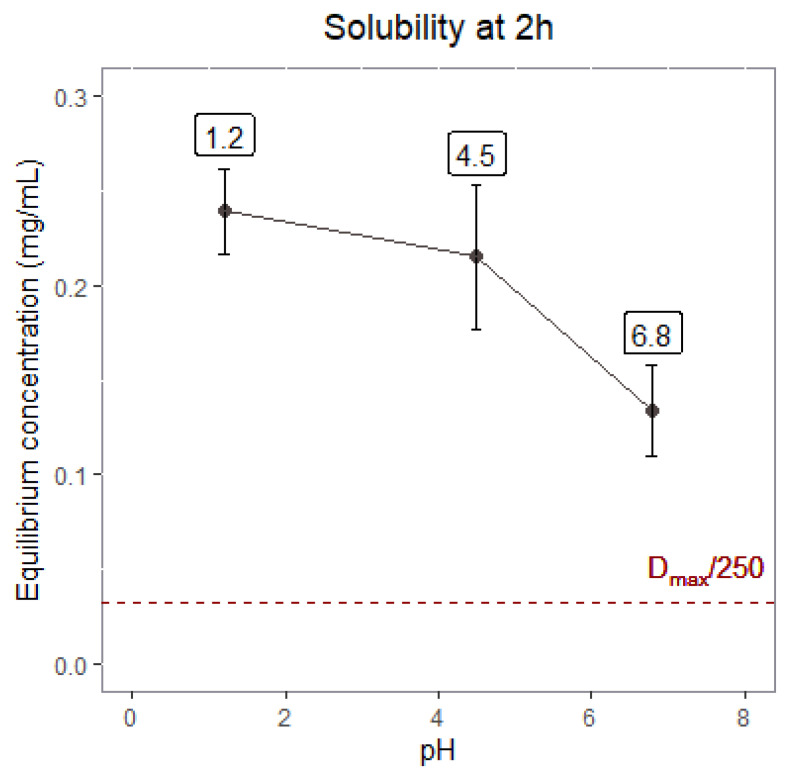
Equilibrium concentrations of silodosin at pHs 1.2, 4.5, and 6.8 after 2 h.

**Figure 3 pharmaceutics-14-02565-f003:**
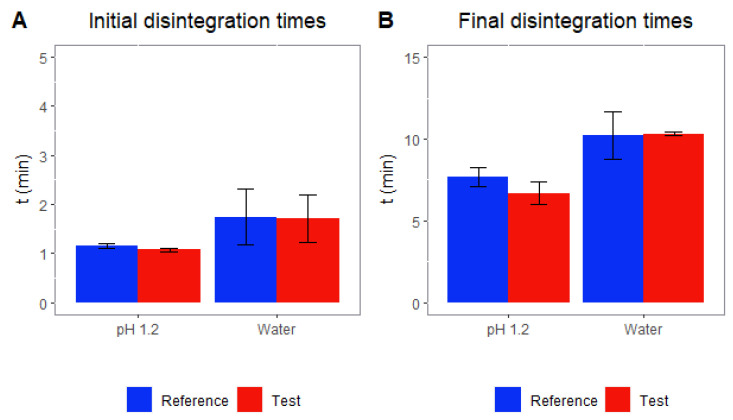
Disintegration times for reference product and test product in water and buffer at pH 1.2.

**Figure 4 pharmaceutics-14-02565-f004:**
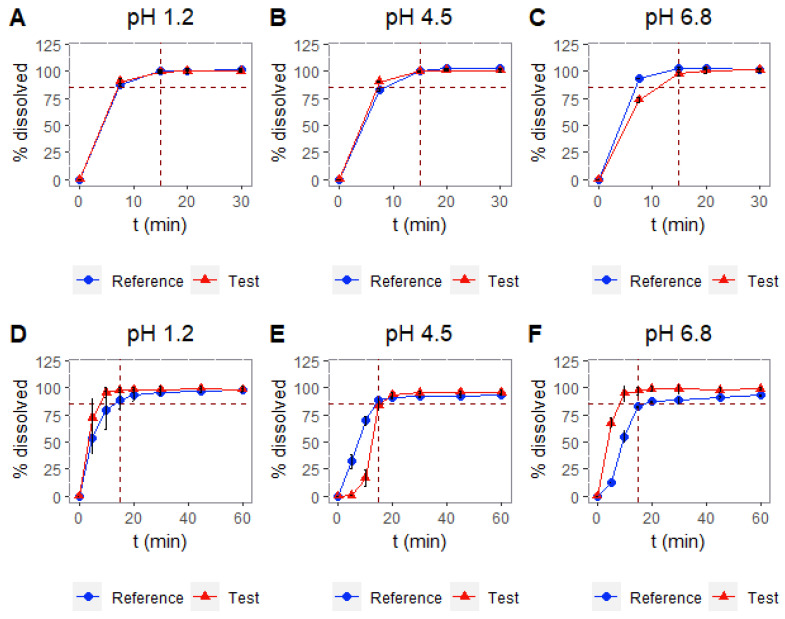
Dissolution profiles for test and reference products at pH 1.2, 4.5, and 6.8. Baskets at 100 rpm (**A**–**C**) and paddles at 50 rpm (**D**–**F**), respectively.

**Figure 5 pharmaceutics-14-02565-f005:**
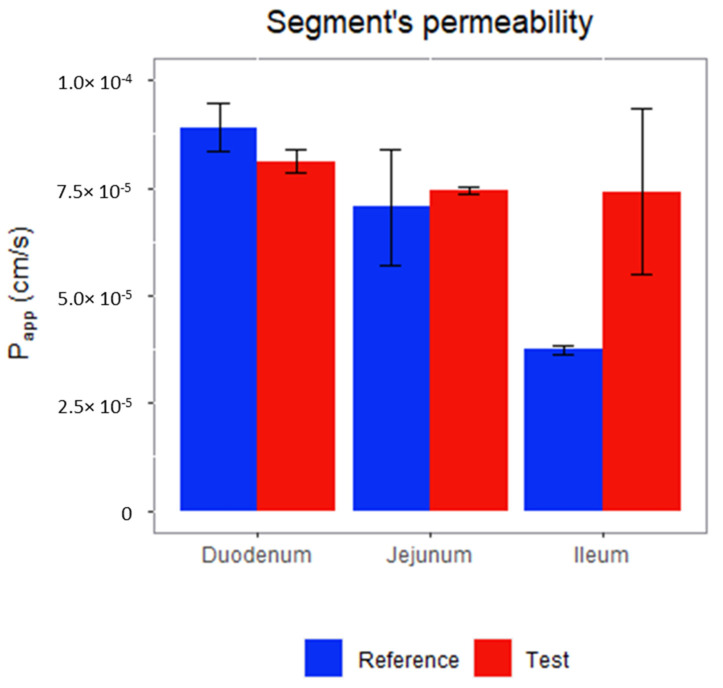
Final product permeability in duodenum, jejunum, and ileum.

**Figure 6 pharmaceutics-14-02565-f006:**
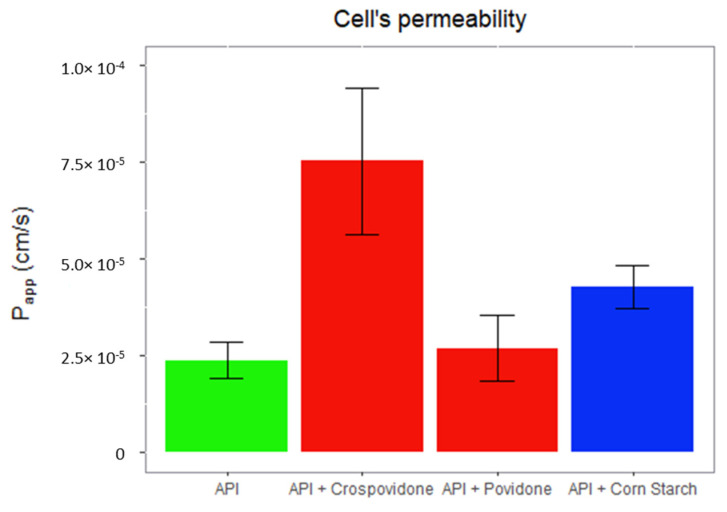
Permeability of silodosin on its own and in presence of excipients.

**Figure 7 pharmaceutics-14-02565-f007:**
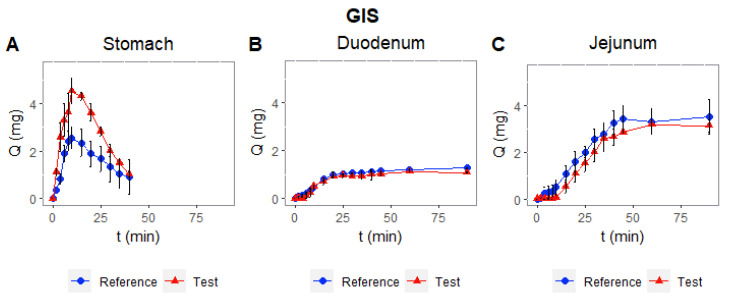
Experimental values of amount dissolved (%) for reference and test silodosin products in each GIS chamber ((**A**)—Stomach, (**B**)—Duodenum, and (**C**)—Jejunum).

**Table 1 pharmaceutics-14-02565-t001:** Qualitative different composition of excipients of the reference and the test products.

Excipients	Reference	Test
Yellow Iron Oxide	✓	
Pregelatinized Corn Starch	✓	
Crospovidone		✓
Povidone		✓

**Table 2 pharmaceutics-14-02565-t002:** In vivo results of the failed bioequivalence study between the test and the reference products.

	Ratio	CI (%)
C_max_	1.12	100.15–126.08
AUC	1.05	95.23–115.09

CI: Confidence interval.

**Table 3 pharmaceutics-14-02565-t003:** Experimental conditions in the GIS for testing the different drug products of the drug.

Fasted-State Test Conditions	GISStomach	GISDuodenum	GISJejunum
Dissolution Media	Simulated Gastric Fluid (SGF), pH 2.0, 0.01 M HCl + 34.2 mM NaCL	Phosphate Buffer 50 mM pH 6.5	/
Initial Volumes	50 mL SGF + 250 mL of water	50 mL	/
Secretions	1 mL/min of SGF	1 mL/min of Phosphate Buffer 100 mM pH 6.5	/

**Table 4 pharmaceutics-14-02565-t004:** Similarity results obtained from the different dissolution experiments for the reference product and the test product.

USP	RPM	pH	Similarity
Baskets	100 rpm	1.2	Similar
		4.5	Similar
		6.8	Similar
Paddles	50 rpm	1.2	Similar
		4.5	No similar (f_2_ = 25.5)
		6.8	No similar (f_2_ = 22.8)

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
