# Peer review of "Exploring a Bioequivalence Failure for Silodosin Products Due to Disintegrant Excipients"

_pharmaceutics, 2022, doi:10.3390/pharmaceutics14122565_

Round 1

Reviewer 1 Report

The manuscript provides great insight into the effect of excipients on the absorption of drugs in the GI tract. My only observation is in figure 1, the data shows only the averages and not the standard deviation. Table 2 shows a difference that makes the test and reference non bioequivalent but the graph shows great similarity. Consider adding the standard deviation to the graph. 

Author Response

Thanks for your revision. All comments has been taken into account. 

final manuscript with highlight changes is attached.

Reviewer 2 Report

The manuscript by González-Álvarez et al., investigated the reasons for a failure of bioequivalence between two formulations of sildospin. The results of this study identifies the change in permeability of silodosin due to difference in excipients used in two different products. This study is interesting and in general this study is well planned. I would recommend accepting this manuscript after a minor revision for the following minor comments.

Line 30: Delete the space in the word ‘d rug’.

Section 2.2.  Move Figure 1 and the results from this section to the results section. Include the Institutional approval statement for in vivo studies. Include error bars to the plasma profiles in figure 1.

Line 109: The section 2.3 heading is missing.

Figure 5: Include the error bars in the dissolution profiles.

Author Response

Thanks for your revision. All comments has been taken into account. 

Reviewer 3 Report

The manuscript is well written, with proper explanation and supporting literature. The work was very well designed and executed. The authors are suggested to make a minor correction. 

  1. Make a detailed note on the sample preparation for In vitro permeability tests.

Author Response

(The authors gave the same response as above.)
